# ABDUCTIVE EXPLANATIONS FOR GROUPS OF SIMILAR SAMPLES

## ABSTRACT

Explaining the decisions of machine learning models is crucial as their use becomes widespread. While many approaches to explanation are based on heuristics or surrogate models without formal guarantees, formal explanations provide reasoning for a particular decision that is guaranteed to be valid. We focus on abductive explanations (AXp) that identify sufficient subsets of input features for a given classification. We extend AXp to not only cover a particular sample, but to cover all of the samples whose features are within a given interval, providing explanations that remain valid even when the features in the explanation vary by up to $\delta$. In addition to applying this notion of *$\delta$-robust AXp* to a single sample, we also consider *group explanations* ($\delta$-gAXp), which give a common explanation for a group of samples that share the same classification. We evaluate our approach by producing explanations for neural networks with the help of Marabou, a neural network verifier. The evaluation shows that, compared to a recent approach for finding a maximally "inflated" explanation, a $\delta$-robust AXp covers a significant volume of the inflated explanation with a dramatically lower runtime. Our evaluation also provides evidence that group explanations capture important features for all the samples within the group much faster than computing explanations for each sample separately.

## 1 INTRODUCTION

Many real world applications are increasingly relying on decision-making systems based on machine learning, including in sensitive contexts such as health care, law enforcement and finance (Karimi et al., 2023). This has spurred calls for a right to explanation for those affected by the decisions of automated decision-making systems. For example, OECD AI Principles recommend that AI systems should "provide information that enable those adversely affected by an AI system to challenge its output"[1] while the data protection regulation of the European Union GDPR requires that users subject to automatic decision-making have the right to "meaningful information about the logic involved"[2]. Indeed, research into methods for explaining machine learning models has blossomed in recent years (Confalonieri et al., 2021; Minh et al., 2022; Bodria et al., 2023; Marques-Silva, 2024).

Many proposed explanation methods, however, can produce misleading explanations (Kumar et al., 2020; Huang & Marques-Silva, 2023a; Marques-Silva & Huang, 2024). Approaches such as the popular SHAP (Lundberg & Lee, 2017) and LIME (Ribeiro et al., 2016) are heuristic and thus do not provide formal guarantees. *Formal* explanations, in contrast, produce verifiably valid explanations (Shih et al., 2018; Ignatiev et al., 2019; Marques-Silva & Ignatiev, 2022). In this paper, we focus specifically on formal *abductive* explanations (AXps) (Marques-Silva & Ignatiev, 2022), which explain why the model classifies a sample as it does. More concretely, an abductive explanation is a subset of the features of an input sample such that it is guaranteed that the classifier assigns the same label to any sample sharing the values of these features with the input sample. AXps, in other words, offer a local feature selection explanation.

Abductive explanations suffer from a notable limitation, in that they are fragile to small perturbations in feature values. An AXp is only valid for the exact feature values of a given sample, so after a

---

minimal perturbation to a single feature it may no longer be valid, even though the perturbed sample is almost identical to the original. This locality can be a limiting factor in real-world applications where measurements are noisy or uncertain. For instance, in medical diagnosis, an explanation for a classification based on a patient's blood pressure being 140.0 mmHg might be invalid for a reading of 140.1 mmHg, although the difference is a slight measurement error. Such a brittle explanation is intuitively inferior to an explanation that holds for all similar values. Furthermore, one might be interested in explanations for a *group* of similar samples. For example, given five patients with similar health records and a similar classification given by the model, it might be more instructive to see explanations for the group than for the individuals.

In this work we consider the extension of AXps to a neighbourhood of samples rather than to a single sample. We introduce $\delta$-robust AXps, where $\delta$ is a robustness bound that ensures that an abductive explanation remains valid when the feature values vary within $\delta$. A separate $\delta_i$ can be set for each feature $i$. With these robustness bounds, we lift AXps from a local to semi-local explanations where explanations are valid for a group of samples. We also introduce a type of group abductive explanations $\delta$-gAXps, which explain groups of similar samples by providing a $\delta$-robust AXps for a representative of the group. When applied to a single sample, our approach is similar to a recent approach to compute *inflated* abductive explanations (Izza et al., 2024b; 2025), which give a maximal interval for each feature within which an AXp is valid. In contrast, our approach first sets a desired interval and then finds a valid AXp for this potentially smaller set of intervals, hugely reducing the computational cost. Given the difficulty of producing formal explanations, it is crucial to find the most efficient methods for obtaining the desired results.

We make the following contributions. We introduce $\delta$-robust AXps, an extension of abductive explanations that provide formal guarantees while being robust to small perturbations, and $\delta$-gAXp, a formalization of abductive explanations for groups of samples. We implement our approach using a state-of-the-art neural network verifier Marabou (Wu et al., 2024), leveraging the formal guarantees and computational efficiency of constraint-solving. We show in a comprehensive empirical evaluation that our approach scales effectively, adding little overhead compared to computing standard AXps. We show that, compared to (maximally) inflated AXps, $\delta$-robust AXps cover a significant portion of the feature space at a fraction of the computational cost. Finally, we evaluate $\delta$-gAXps based on their induced formal feature attribution scores (FFA) (Yu et al., 2023) and show that they effectively capture the feature importance within a group. Using the FFA scores of each individual sample of the group as the ground truth, $\delta$-gAXps perform better than SHAP and LIME, and comparable to a brute-force method of computing the average of the FFAs for all samples of the group, with an orders of magnitude lower runtime.

## 2 RELATED WORK

Explainable AI has gained significant attention in recent years (Confalonieri et al., 2021; Minh et al., 2022; Bodria et al., 2023), leading to the development of diverse methodologies including heuristic explanations, surrogate models, feature attribution, feature selection, and interpretable models. In this work, we focus on formal explanations, especially feature selection.

Researchers have approached formal explanations from various perspectives. Among these, feature selection methods, which identify subsets of features that constitute explanations, can be roughly divided into *abductive* and *counterfactual* explanations (Wachter et al., 2018; Shih et al., 2018; Ignatiev et al., 2019; Izza & Marques-Silva, 2021; Ignatiev & Marques-Silva, 2021; Parmentier & Vidal, 2021; Wu et al., 2023b; 2024; Karimi et al., 2020; Audemard et al., 2023; Marques-Silva & Ignatiev, 2022; Marques-Silva, 2024; Karimi et al., 2023; Guidotti, 2024). Intuitively, abductive explanations answer the question *why this classification* and counterfactual explanations answer *how to obtain a different classification*. Formally, an abductive explanation is a subset of a sample's features such that any sample fixing these features to their respective values receives the same classification as the original sample. A counterfactual explanation, conversely, is a subset of features such that any sample with different values for these features receives a different classification than the original sample.

Robust explanations have been studied for counterfactual explanations (Jiang et al., 2023; 2024), but they have received less attention for AXps. For AXps, a type of robustness called distance-restricted AXps (Izza et al., 2024a; Huang & Marques-Silva, 2023b) and $\epsilon$-robust explanations (Wu

et al., 2023a) has been considered, where the *relevant* features, i.e. features that are part of the explanation, must remain fixed while the *irrelevant* features, i.e. features not in the explanation, can only vary within a given bound $\epsilon$.

In contrast to distance-restricted explanations, our $\delta$-robust AXps extend AXps by remaining valid when the relevant features vary by up to $\delta$, regardless of how much the irrelevant features vary. In the parlance of Jiang et al. (2024), our $\delta$-robust AXps are robust against noisy execution. As mentioned above, the closest to this sense of robustness for AXps are inflated AXps (Izza et al., 2024b), which give bounds to the relevant features so that explanations remain valid when the input variations occur inside the bounds. The algorithm to generate inflated AXps maximizes the bounds of each feature in a given AXp, one feature at a time, with multiple calls to a solver that checks whether the current explanation is valid. This approach has two drawbacks compared to our $\delta$-robust AXps. First, the feature(s) whose bounds are maximized last can be left with tiny bounds or no bounds at all. Second, they are computationally costly since they require multiple calls to a validity checker.

Group explanations present a separate but related challenge, since they attempt to explain multiple samples with the same classification simultaneously. They have recently gained attention for counterfactual explanations (Warren et al., 2023; Carrizosa et al., 2024a;b; Wielopolski et al., 2024), but, to the best of our knowledge, group explanations have not been considered for AXps. Our work addresses these gaps in the literature by considering *robust abductive* explanations, and applying them to produce abductive explanations for *groups of samples*.

## 3 ABDUCTIVE EXPLANATIONS

Let $f : \mathbb{R}^d \to \{1, \ldots, k\}$ be a classifier that maps samples $\mathbf{x} = \{x_1, \ldots, x_d\} \in \mathbb{R}^d$ to classes $c \in \{1, \ldots, k\}$. An abductive explanation (AXp) answers the question *why does $f$ classify $\mathbf{x}$ as $c$* by identifying a subset of features $E$, such that the classifier assigns the label $c$ to any sample that shares all values of $E$ with $\mathbf{x}$. Formally, a subset of features $E \subseteq \mathbb{F}$ is an AXp of $\mathbf{x}$ if

$$\forall \mathbf{z} \in \mathbb{R}^d : \bigwedge_{i \in E} (z_i = x_i) \implies f(\mathbf{z}) = f(\mathbf{x})$$

and there is no proper subset of $E$ satisfying this condition.

### 3.1 $\delta$-ROBUST ABDUCTIVE EXPLANATIONS

A shortcoming of an abductive explanation (AXp) (Ignatiev et al., 2019) is that a small change to one of the relevant features (i.e. a feature in the explanation) can make the explanation invalid. In many real world scenarios, measurement errors can occur, so the fragility of formal explanations limits their practicality. To address this limitation, we propose a variant of abductive explanations that is valid when relevant features vary within specified bounds. This new formulation not only makes abductive explanations robust to input variation, but it allows us to define formal explanations for groups of samples, clarifying the behaviour of classifiers on a group of samples rather than just one.

A $\delta$-robust AXp for a sample $\mathbf{x}$ with classification $f(\mathbf{x}) = c$ is a subset of features $E \subseteq \mathbb{F}$ and a set of robustness bounds $\{[x_i - \delta_i, x_i + \delta_i] \mid i \in E\}$, such that $f(\mathbf{z}) = c$ for any input $\mathbf{z} \in \mathbb{R}^d$ where $z_i \in [x_i - \delta_i, x_i + \delta_i]$ for all $i \in E$. The robustness parameter $\boldsymbol{\delta} = \{\delta_i\}_{i \in E}$ controls the maximum allowed variation in each relevant feature. Formally, a pair $(E, \boldsymbol{\delta})$ is a $\delta$-robust AXp if

$$\forall \mathbf{z} \in \mathbb{R}^d : \bigwedge_{i \in E} (z_i \in [x_i - \delta_i, x_i + \delta_i]) \implies f(\mathbf{z}) = f(\mathbf{x})$$

and there is no proper subset of $E$ satisfying this condition.

This definition ensures that abductive explanations remain valid when relevant features vary within their specified bounds, addressing the fragility to input variation of standard abductive explanations. For convenience, when $\delta_i = v$ for each $i \in E$, we may write that $\delta = v$ and refer to a $v$-robust AXp. Note that $\delta$-robust AXps are a generalization of AXps. If $(E, \boldsymbol{\delta})$ is a $\delta$-robust AXp for a sample $\mathbf{x}$, then $E$ is an AXp for $\mathbf{x}$. Similarly, an AXp is a 0-robust AXp.

**Example 1.** *Figure 1 illustrates an AXp and a $\delta$-robust AXp with $\delta = 0.09$. In both, the irrelevant features are the same and can take whatever value without changing the classification. In the relevant features, however, the AXp shows a precise point where the explanation is valid, represented*

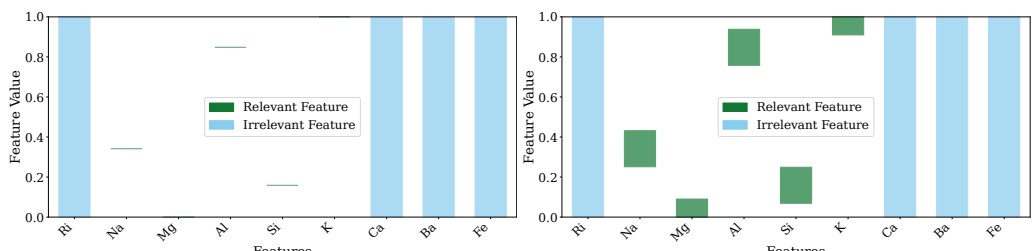

Figure 1: Comparison of two explanations for the first sample with class 5 of the Glass dataset. An AXp (left) and a $\delta$-robust AXp for $\delta = 0.09$ (right).

*by the narrow green line corresponding to the feature values of the input sample, while the $\delta$-robust AXp shows intervals within which the relevant features can take any value without the classification changing, represented by the large green rectangles corresponding to the intervals of validity.*

To compute a $\delta$-robust AXp, we assume that features are normalized to $[0, 1]$ and that we have a formal model of the classifier in question $f$ as a set of logical constraints and an oracle for checking whether a given set of constraints is satisfiable. In our implementation we use Marabou, a formal verifier of properties in neural networks based on constraint solving (Wu et al., 2024). We impose further constraints as the formula $\phi$ during the course of the algorithm. The algorithm takes as input a sample $\mathbf{x}$ and a set of robustness bounds $\boldsymbol{\delta} = \{\delta_i\}_{i=1}^d$, and finds a set $E$ such that $(E, \{\delta_i\}_{i \in E})$ is a $\delta$-robust AXp for $\mathbf{x}$.

In Algorithm 1, we modify the standard deletion-based algorithm for computing minimal unsatisfiable sets of constraints (Chinneck & Dravnieks, 1991) to compute a $\delta$-robust AXp. First, if the classification is not guaranteed even with all features fixed to be within their respective bounds, there are no $\delta$-robust AXps and the algorithm terminates (lines 2–3). Otherwise, include all features $\mathbb{F}$ are included in an over-approximation of a $\delta$-robust AXp E, and each feature $i$ is considered in turn to determine whether to include it in E (lines 5–9). The order of iteration can be arbitrary, but some orders might result in smaller or otherwise more desirable explanations (Wu et al., 2023a); we discuss this more in the Appendix. For each feature $i$, the algorithm sets constraints so that $i$ can take any value, and other features $j$ can take values based on whether it is in $E$ or already deemed irrelevant. The features in $E$ are the potentially relevant features, so they must be within their robustness bounds (line 6). An exception is the feature $i$, which we allow to take any value in order to check whether it is irrelevant (lines 6–7). Features not in $E$ are irrelevant features, so they can take any value in $[0, 1]$ (line 7). In line 8 the function GUARANTEED($\phi, f(\mathbf{z}) = f(\mathbf{x})$) uses a constraint-solving oracle and returns `True` iff $f(\mathbf{z}) = f(\mathbf{x})$ holds under all values of $\mathbf{z}$ that satisfy the constraints $\phi$. If this is the case, then the feature $i$ can demonstrably take any value without changing the classification, so $i$ is irrelevant and it is removed from $E$ (line 9).

### 3.1.1 COMPARISON TO INFLATED ABDUCTIVE EXPLANATIONS

Inflated abductive explanations (Izza et al., 2024b) are a related approach to our $\delta$-robust AXps. They maximize the bounds of relevant features one feature at a time and separately for the lower and upper

---

**Algorithm 1** Computing a $\delta$-robust AXp for sample $\mathbf{x}$

1: **function** COMPUTEROBUSTAXP($\mathbf{x}, \boldsymbol{\delta}$)
2:      **if** $\neg$GUARANTEED($\{(|z_j - x_j| \le \delta_j) \mid j \in \mathbb{F}\}, f(\mathbf{z}) = f(\mathbf{x})$) **then**
3:          **return** `False`
4:      $E \leftarrow \mathbb{F}$
5:      **for** each feature $i$ in $\mathbb{F}$ **do**
6:          $\phi \leftarrow \{(|z_j - x_j| \le \delta_j) \mid j \in E \setminus \{i\}\}$
7:          $\phi \leftarrow \phi \cup \{(z_j \in [0, 1]) \mid j \in \mathbb{F} \setminus (E \setminus \{i\})\}$
8:          **if** GUARANTEED($\phi, f(\mathbf{z}) = f(\mathbf{x})$) **then**
9:             $E \leftarrow E \setminus \{i\}$
10:      **return** $E$

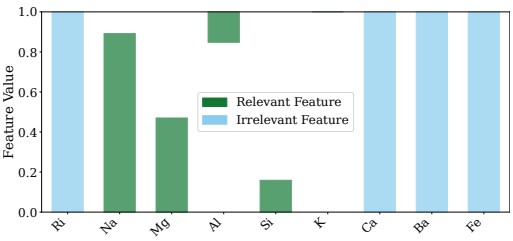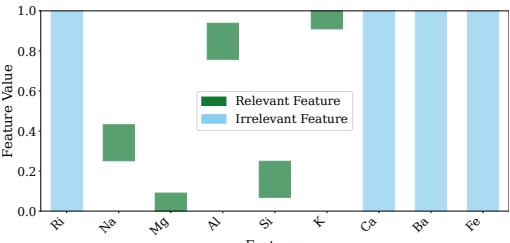

Figure 2: Comparison of two explanations for the first sample with class 5 of the Glass dataset. An inflated AXp (left) generated by inflating a $\delta$-robust AXp for $\delta = 0.09$ (right).

bound. They achieve this by first computing an AXp and then considering each feature $i$ in the AXp in turn. They increase the bound $b_i$ for feature $i$ by a small, predetermined amount $\eta$ (precision) and check whether the explanation is still valid with a call to an oracle (typically a solver for a constraint language such as propositional satisfiability or mixed-integer linear programming). When the explanation is no longer valid, they take the previous bound (i.e. $b_i - \eta$) and repeat the process for the lower bound of feature $i$. The result is an explanation together with a set of maximal bounds on each relevant feature that guarantee that varying the values of the relevant feature within these bounds does not change the classification. In other words, inflated AXps are a similar to $\delta$-robust AXps with a separate $\delta$ for each feature and each direction (i.e. increasing the value and decreasing the value) where all the $\delta$s are maximal. Computing an inflated AXp is more computationally demanding than computing a $\delta$-robust AXp, since to inflate, an oracle needs to be called potentially hundreds of times. Additionally, the features considered last in computing an inflated AXp might get very low bounds.

**Example 2.** *In Figure 2 we illustrate an inflated AXp and a $\delta$-robust AXp sharing the same underlying AXp. The bounds of the inflated AXp are larger in general, but since inflated AXps are maximized one feature at a time, some features have smaller bounds. Notably, the bounds for feature 'K' could not be inflated at all and instead the feature must remain fixed to a single value in the inflated AXp. Conversely, the $\delta$-robust AXp guarantees given bounds for each feature, so every feature has flexibility. This $\delta$-robust AXp covers 43% of the area of the inflated AXp.*

### 3.2 GROUP EXPLANATIONS

While AXps explain the classification of a single sample, in many cases it is insightful to explain the classification of a group of samples. In medical diagnosis, for example, researchers might want to understand what features are important to diagnose a condition in general, not just for a particular patient. Group explanations address this need by identifying the features that explain why the samples in a group receive the same classification. Group explanations can be thought of as semi-local, since they do not apply to a single sample (local) or to the whole model (global), but to a group of samples. They are most effective with samples that are naturally grouped together, such as patients with similar symptoms, customers with similar profiles, or products with similar characteristics.

In Algorithm 2, we illustrate how to compute a type of group explanation, $\delta$-gAXp, by computing $\delta$-robust AXps on a representative of the group with robustness bounds that depend on the variance of feature values occurring in the group. The algorithm takes a group of samples $X = \{\mathbf{x}_1, \ldots, \mathbf{x}_n\}$ that share the same classification and a vector $\boldsymbol{\delta}$ as input. First, it computes a representative of the group by averaging the samples in the group (line 2). If the representative receives a different clas-

---

**Algorithm 2** Computing a $\delta$-gAXp for a group of samples with the same classification

1: **function** COMPUTEGROUPAXP($X = \{\mathbf{x}_1, \ldots, \mathbf{x}_n\}, \boldsymbol{\delta}$)
2:     $\bar{\mathbf{x}} \leftarrow \text{Mean}(X)$
3:     **if** $f(\bar{\mathbf{x}}) \neq f(\mathbf{x}_1)$ **then return** False
4:     $\boldsymbol{\gamma} \leftarrow \boldsymbol{\delta} \cdot \text{StandardDeviation}(X)$
5:     $E \leftarrow \text{COMPUTEROBUSTAXP}(\mathbf{x}, \boldsymbol{\gamma})$
6:     **return** $(E, \boldsymbol{\gamma})$

---

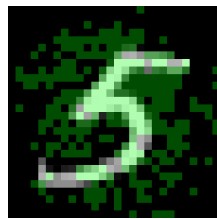 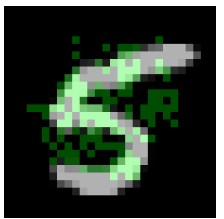 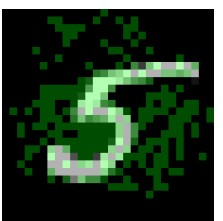 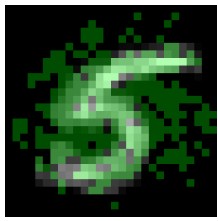

Figure 3: From left to right, the first three images represent individual AXps for three images in MNIST, while the last image represents a $\delta$-gAXp for $\delta = 0.3$ for the former images. The green pixels represent the relevant features in the explanations.

sification than the samples in $X$, the group is too dissimilar and has no $\delta$-gAXp (line 3). Otherwise, the robustness bounds of $\delta_i$ are computed so that they are proportional to the standard deviation of each feature $i$ within the group for the representative (line 4). Finally, it checks whether a $\delta$-robust AXp exists with Algorithm 1 using these bounds (line 5). If the $\delta$-robust AXp exists, we call the output a $\delta$-gAXp $= (E, \boldsymbol{\gamma})$. For any sample $\mathbf{z} = \{z_1, \ldots, z_d\}$ it holds that if $|z_i - \bar{x}_i| \leq \gamma_i$ for every feature $i \in E$, then $f(\mathbf{z}) = f(\bar{\mathbf{x}})$.

The parameter $\boldsymbol{\delta}$ multiplies the standard deviation of the features in the group producing the final bounds $\boldsymbol{\gamma}$ for the features. This formulation sets wider bounds for features that have diverse values in the group and narrower bounds for features that have similar values in the group. If $\delta = 0$, the $\delta$-gAXp is a AXp for the group representative, while larger values of $\delta$ increase the robustness bounds, making the explanation valid for a wider range of samples.

**Example 3.** *In Figure 3, we compute an AXp for three MNIST samples and their common $\delta$-gAXp (for $\delta = 0.3$). The individual explanations show different patterns for each sample, reflecting their specific characteristics, while the $\delta$-gAXp captures the common features that are important across the three samples, reflecting the general characteristics that make them belong to the same class.*

## 4 EMPIRICAL EVALUATION

We conducted an empirical evaluation using 2x12 core Xeon E5 2680 v3 2.50GHz CPUs with 128GB DDR4-2133 of RAM. The code for the experiments can be found in the supplementary material and will be made available in open source.

**Datasets**  We evaluate our methods on both *classic machine learning datasets* Glass (OpenML, a), Leaf (OpenML, b), Parkinsons (OpenML, c), Diabetes (Efron et al., 2004) and Wine (Aeberhard & Forina, 1992) and *synthetic datasets* generated using scikit-learn (function make_classification). The classic datasets represent real-world scenarios with varying dimensionality and complexity, while the synthetic datasets allow us to control dimensionality and feature characteristics for scalability analysis. In the synthetic datasets, we have 10000 samples, 2 classes, and 10 to 25 features. The make_classification function generates datasets with informative and noisy features. Informative features are those that contribute to the target variable, while noisy features are random and do not provide useful information for classification. We set the number of informative features to $\lfloor n/2 \rfloor$, where $n$ is the total number of features.

**Machine Learning Model**  We train a fully-connected neural network (NN) with 2 hidden layers of 10 neurons each and ReLU as their activation function for each dataset. We train each NN for 500 epochs with a learning rate of 0.001, a batch size of 16, and a patience of 10. We use 0 as the seed. We compute explanations with Marabou (Wu et al., 2024), which encodes the NN into a logical formula and allows for formally verifying properties of the NN.

**Comparisons**  We compare $\delta$-robust AXps against inflated AXps (Izza et al., 2024b) (computed with a reimplementation using Marabou) as the primary baseline, since they represent the most relevant alternative approach. We evaluate the total volume of feature values covered by both approaches. For $\delta$-gAXps, i.e. group explanations, we compare the formal feature attribution (FFA) (Yu et al., 2023) induced by our approach against SHAP, LIME, and a brute-force approach of

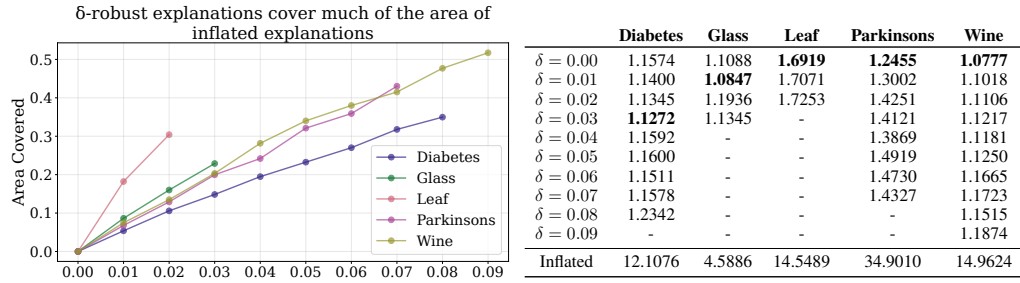

| | Diabetes | Glass | Leaf | Parkinsons | Wine |
|---|---|---|---|---|---|
| $\delta = 0.00$ | 1.1574 | 1.1088 | **1.6919** | **1.2455** | **1.0777** |
| $\delta = 0.01$ | 1.1400 | **1.0847** | 1.7071 | 1.3002 | 1.1018 |
| $\delta = 0.02$ | 1.1345 | 1.1936 | 1.7253 | 1.4251 | 1.1106 |
| $\delta = 0.03$ | **1.1272** | 1.1345 | - | 1.4121 | 1.1217 |
| $\delta = 0.04$ | 1.1592 | - | - | 1.3869 | 1.1181 |
| $\delta = 0.05$ | 1.1600 | - | - | 1.4919 | 1.1250 |
| $\delta = 0.06$ | 1.1511 | - | - | 1.4730 | 1.1665 |
| $\delta = 0.07$ | 1.1578 | - | - | 1.4327 | 1.1723 |
| $\delta = 0.08$ | 1.2342 | - | - | - | 1.1515 |
| $\delta = 0.09$ | - | - | - | - | 1.1874 |
| Inflated | 12.1076 | 4.5886 | 14.5489 | 34.9010 | 14.9624 |

Figure 4: Across 100 samples, average area of inflated AXps covered by $\delta$-robust AXps (left) and average runtime (right).

computing explanations separately for each sample in the group and aggregating them afterwards. We consider the FFA scores of the individual samples as the ground-truth. We also evaluate the runtime since all the explanation problems we consider are computationally hard.

### 4.1 EVALUATION OF $\delta$-ROBUST EXPLANATIONS

We evaluate our approach across 100 samples by comparing the volume of a $\delta$-robust AXp to the volume of an inflated AXp based on the same underlying AXp (i.e. sharing the same relevant features), and by comparing their runtimes. We use a precision of $\eta = 0.01$ for the inflated AXp. To present cases where applications could realistically use $\delta$-robust AXps, we only show data for values of $\delta$ where at least 50% of the samples have a $\delta$-robust AXp.

Figure 4 shows the results. On the left side we see that $\delta$-robust AXps achieve up to 20–50% of the coverage of inflated AXps, and on the right that they are 4 to 30 times faster to compute than inflated AXps. Thus, even though $\delta$-robust AXps do not maximize the validity bounds of each feature, they provide valid explanations covering a substantial percentage of the area of inflated AXps at a fraction of the computational cost. Therefore, $\delta$-robust AXps offer an attractive trade-off between runtime and the size of the bounds. This might be appealing in real-time decision support systems or in applications with limited computational resources, where they aim to find explanations for a large, but not necessarily maximal, feature intervals. Note that the coverage of $\delta$-robust AXps increases monotonically with $\delta$ across all datasets without any computational overhead. One can thus disregard the computational cost in choosing $\delta$ and instead choose it according to the needs of the application in question, or to maximize the are covered subject to there still existing AXps.

### 4.2 EVALUATION OF GROUP EXPLANATIONS

To the best of our knowledge, there is no standard dataset or approach to evaluate group abductive explanations, so we evaluate the quality of $\delta$-gAXps with the so-called weighted FFA (Yu et al., 2023), which has been introduced to offer a formally guaranteed alternative to heuristic feature attribution methods, such as SHAP. The weighted FFA score of a feature for a sample is, intuitively, the proportion of AXps in which this feature occurs in, weighted so that shorter explanations have additional importance. To compute the FFA, we generate all AXps with the MARCO algorithm (Liffiton et al., 2015; Ignatiev & Marques-Silva, 2021) and compute the weighted FFA (Yu et al., 2023). We compare the FFA scores generated using $\delta$-gAXps to the FFA scores from each individual sample in the group, and evaluate how close they are.

To put the result in context, we provide the same evaluation for three further approaches: "Aggregated group FFA", SHAP, and LIME. The Aggregated group FFA is the brute-force approach of computing all AXps for each sample separately and taking the mean of the FFA score for each feature. As metrics, we first use the *feature importance error*, which is the average normalized mean absolute error between the group-FFA and the FFA of each sample in the group. We provide a scale-invariant measure of accuracy by normalizing the mean absolute error by the mean of the target feature importance values. Secondly, we evaluate the *feature importance correlation* as the average Spearman correlation between the ranking induced by the group-FFA and the ranking induced by

| | Diabetes | | Glass | | Leaf | | Parkinsons | | Wine | | Synthetic | |
| | Err. | Cor. | Err. | Cor. | Err. | Cor. | Err. | Cor. | Err. | Cor. | Err. | Cor. |
|---|---|---|---|---|---|---|---|---|---|---|---|---|
| Aggr. | **.232** | **.727** | **.202** | .563 | **.095** | .817 | **.109** | .907 | **.114** | **.852** | **.176** | **.746** |
| LIME | – | .493 | – | .164 | – | .200 | – | .482 | – | .586 | – | .529 |
| SHAP | – | .588 | – | .355 | – | .374 | – | .592 | – | .525 | – | .634 |
| $\delta = .00$ | .277 | .684 | .279 | .712 | **.098** | **.855** | .117 | .905 | **.130** | **.824** | **.239** | **.670** |
| $\delta = .02$ | **.271** | .699 | .288 | .712 | .105 | .840 | **.116** | .907 | .133 | .818 | .240 | .661 |
| $\delta = .04$ | .272 | .716 | .277 | .706 | .110 | .828 | .117 | .906 | .135 | .813 | .246 | .657 |
| $\delta = .06$ | .273 | **.723** | .280 | .725 | .106 | .832 | .118 | .906 | .135 | .811 | .250 | .659 |
| $\delta = .08$ | .274 | .712 | .271 | **.746** | .105 | .837 | .118 | .906 | .135 | .810 | .254 | .664 |
| $\delta = .10$ | .276 | .708 | .258 | .738 | .106 | .827 | .119 | .907 | .138 | .817 | .256 | .664 |
| $\delta = .12$ | .292 | .695 | .258 | .738 | .107 | .826 | .119 | .907 | .138 | .819 | .257 | .655 |
| $\delta = .14$ | .284 | .720 | **.250** | .739 | .111 | .828 | .121 | .908 | .134 | .820 | .259 | .655 |
| $\delta = .16$ | .286 | .702 | .253 | .737 | .112 | .829 | .117 | **.910** | .134 | .817 | .260 | .652 |
| $\delta = .18$ | .278 | .701 | .265 | .729 | .111 | .818 | .121 | .902 | .137 | .821 | .261 | .640 |

Table 1: Average feature importance error (Err.) and ranking correlation (Cor.) across 10 different groups for aggregated explanations (Aggr.), SHAP, LIME, and $\delta$-gAXps for $\delta \in \{.00, .02, ..., .18\}$. We denote in bold the best scores overall and among $\delta$-gAXps.

the FFA of each sample in the group. We compute the ranking by ordering features according to their FFA score. We also compare our approach to the heuristic explanation methods SHAP and LIME by ordering the features in the order of importance given by SHAP and LIME and computing the correlation between their order and the ranking induced by the FFA of each sample. For SHAP and LIME, we do not compare the feature importance scores given by their FFA since the scores themselves are not commensurable.

In Table 1 we see that $\delta$-gAXps are more accurate than SHAP and LIME. The FFA from $\delta$-gAXps have at least 10 percentage points higher correlation to the ground-truth than SHAP on every dataset except the synthetic, and on some datasets more than 30 percentage points higher, while they have 10 to 40 percentage points higher correlation than LIME. The brute-force aggregation approach generally produces a lower error and higher correlation than $\delta$-gAXp, but in most cases not by much. In some cases $\delta$-gAXps produce *higher* correlation than the aggregation approach, which is notable since the aggregation considers (and thus has to compute) all explanations for all samples separately, while computing $\delta$-gAXps is much more tractable.

Figure 5 illustrates the computational advantage of $\delta$-gAXps. We report three runtime measurements. 'Individual' refers to the mean runtime for computing all AXps for each individual sample, 'Group' refers to the mean runtime for computing all $\delta$-gAXps across all values of $\delta$ (as listed in Table 1), and 'Aggregated' refers to the mean runtime for the naive aggregation method that computes all AXps for all samples in a group. In the plot on the left, we see that $\delta$-gAXps are an order of magnitude faster to compute than aggregated explanations, and that, as the number of features increases, the runtime of the naive aggregation method increases faster than the runtime of $\delta$-gAXps. We also see that the runtime of $\delta$-gAXps is similar to the runtime of computing all AXps for a single sample (compare Group to Individual). In the plot on the right, we see that the runtime of $\delta$-gAXps remains constant as the group size increases, while the runtime of aggregated explanations increases linearly. Regarding the runtime of computing AXps for individual samples (right side of Figure 5), the mean runtime is 183 seconds and the standard deviation is 1384 seconds (note that these are independent of the group size). This points to another benefit of $\delta$-gAXps: they seem to avoid some outliers with a huge number of AXps that make computing all AXps for some samples much slower, as can be seen by the fact that computing all $\delta$-gAXps takes well below 100 seconds for each group size.

**Summary of the evaluation** $\delta$-robust AXps achieve up to 20–50% of the coverage of inflated abductive explanations and are 4 to 30 times faster to compute. The FFA scores induced by $\delta$-gAXps are comparable to (and sometimes better than) the FFA scores induced by the aggregated method, while being an order of magnitude faster to compute. Furthermore, the runtime of $\delta$-gAXps is independent of the size of the groups of samples, and it scales similarly to the runtime of individual explanations with respect to dataset dimensionality.

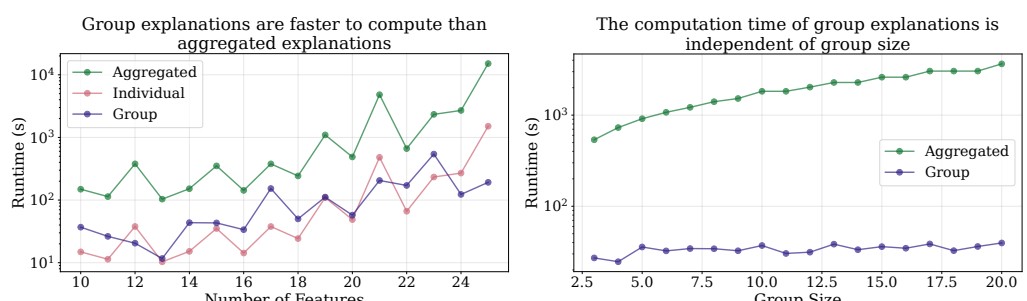

Figure 5: Mean time (s) to generate explanations for (left) groups of 10 samples on datasets with different numbers of features (10–25) and (right) groups of different sizes (3–20 samples) on datasets with 10 features. Each point represents the average of 25 instances (5 groups · 5 datasets).

### 4.3 LIMITATIONS

A limitation of $\delta$-robust AXps is a trade-off between $\delta$ and the number of explanations: there might be only a few or no explanations at all, especially with larger values for $\delta$. For example, in Figure 4 (left) less than 50% of samples have an explanation for some datasets with $\delta = 0.03$. Additionally, the possible bounds for different features might be lopsided, and in that case finding close to maximal values for $\delta$ might be difficult and one would need to revert to computing inflated AXps.

A limitation of $\delta$-gAXps is that, in our experiments, $\delta$-gAXps are not formally valid for each individual sample in the group. This is because the features within a group vary by more than the values of $\delta$ under which some formal explanations can still be found. This might limit the usefulness of $\delta$-gAXps in some applications, although our evaluation based on FFA values shows that the given explanations correctly identify features that are important for the obtained classification of the samples in a group. To overcome this, one could divide groups into smaller subgroups of more similar samples, or even combining $\delta$-robust AXps with a distance restriction on the irrelevant features (Izza et al., 2024a), since the lower the distance restriction, the easier it is to find an explanation.

Finally, in our evaluation we compute a single inflated AXp among multiple possible ones, so an inflated AXp with more volume might exist. Very recently it was shown that computing a maximum-size inflated AXp is much more computationally expensive than computing an arbitrary inflated AXp (Izza et al., 2025), so $\delta$-robust AXps would cover less area compared to a maximum inflated AXp, but they would be even faster.

## 5 CONCLUSION

We introduced $\delta$-robust AXps, extending robustness analysis from counterfactual to abductive explanations in a way that complements existing robustness notions for abductive explanations such as inflated and distance-restricted abductive explanations. The notion of $\delta$-robust AXp defines formal abductive explanations that are robust to input variations, making them more practical for real-world applications with measurement noise, such as healthcare, finance, and autonomous systems. Furthermore, we introduced $\delta$-gAXps, semi-local explanations that explain groups of samples instead of individual samples. Our empirical evaluation demonstrated that $\delta$-robust AXps achieve 20–50% of the area coverage of inflated abductive explanations while requiring 4 to 30 times less runtime. We also demonstrated that $\delta$-gAXps capture commonly important features within groups of samples, achieving an accuracy comparable to a brute-force method that aggregates all explanations for all samples in the group separately, but at a fraction of the computational cost.

### REPRODUCIBILITY STATEMENT

All code related to the presented experiments is provided in the supplementary material with a README file describing how to rerun the experiments.

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

## A  APPENDIX

(Wu et al., 2023a) proposed a sensitivity-based ordering to compute small AXps. The idea of their approach is to sort features by how much they influence the output of the classifier. The feature that influences the output the least is the first feature in the ordering. This approach leads to small explanations since it is more probable that features to which the classifier is not sensitive are irrelevant and can thus be removed from the over-approximation of an AXp. We experiment with a smoothed version of the sensitivity of a feature by averaging it with the sensitivity of its neighbors, in order to produce explanations that include pixels that are close to each other. We call our approach smoothed sensitivity ordering. In Figure 6 we can see the explanations generated using different orderings. The VeriX ordering produces explanations that are small but are unintuitive, while the smoothed sensitivity ordering produces explanations that are concentrated towards the center and capture the shape of the digit better.

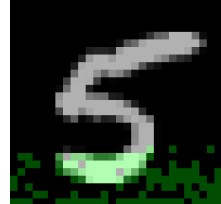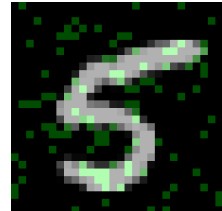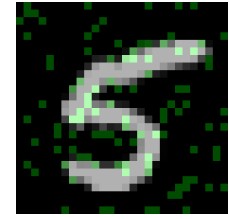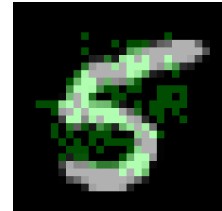

Figure 6: The first explanation generated using different orderings. From left to right, the orderings are: linear, random, VeriX, and smoothed sensitivity. The green pixels represent the relevant features in the explanations.

