# OpenReview forum: "Abductive Explanations for Groups of Similar Samples"
_ICLR.cc/2026/Conference — ICLR 2026 Conference Withdrawn Submission_

### Official Review · Reviewer_VYDF · 2025-10-20

**Soundness:** 4
**Presentation:** 3
**Contribution:** 1
**Rating:** 4
**Confidence:** 4

**Summary:**

The paper investigates formal abductive explanations for similar inputs,
where one restricts the perturbation of relevant features to some bounded neighborhood,
and irrelevant features can take on any value.
This group version is based on the single-input version, where a group representative is selected.

**Strengths:**

- The paper is well-written, streamlined, and easy to understand.
- It considers an important problem of obtaining formal explanations, especially for groups instead of individual inputs.
- All assumptions of the proposed approach are clearly stated, and the problem statement is defined.
- The running example/illustrations help with understanding the approach.
- The evaluation follows scientific practice.

**Weaknesses:**

- The paper is solid, but has little novelty: The proposed approach is very similar to the inflated version, but also to the distance-restricted version mentioned in the related work section. The difference is not algorithmically, only in how you select the perturbation setting.
- Running a single verification query (for a fixed delta) compared to an iterative procedure per feature is trivially faster, but has to be repeated for different deltas.
- While I find the group explanations the interesting part of the paper, the proposed approach potentially has some flaws (see questions below), and does not provide formal guarantees on the group as the $\delta$ is only based on the (scaled) standard deviation, which might not cover all samples of the group.
- Evaluation was done on small networks (2 layer, 10 neurons) and could be enhanced using more complex datasets (MNIST).

**Questions:**

- How does one decide on the perturbation setting and the delta in practice?
- What exactly is meant by "area covered"? E.g., in Fig. 2: Do you compute the area of each green box, sum them up, and then divide it by the sum of the other approach? Eyeballing the figure, the 43% coverage mentioned in Example 2 looks like it. However, I don't think this is an adequate comparison, as the area does not directly relate to how much of the input domain is covered: The volume of the combined perturbation is much, much larger for the inflated version. To demonstrate it more clearly, if each rectangle covers 1/2 of the inflated version, then your area coverage would say 50% coverage. But the volume is only 1/2^n with n features.
- Why does the computational overhead stay constant for different deltas? Usually, verification queries get harder with a larger delta. Would you expect it to change with different network sizes?
- For the inflated version, wouldn't it be possible to run a binary search instead of a small fixed additive value? This would drastically speed up that version, rendering the benefit of the proposed approach as less significant.
- Why didn't you run the approach on MNIST in the evaluation (wondering as you used MNIST in e.g. Fig. 3)?
- For the group evaluation in Tab. 1: Are the mentioned $\delta$s the bold $\delta$ in Alg. 2, i.e., you only select e.g. 2% of the standard deviation? If not, how do they relate?

---

### Official Review · Reviewer_R313 · 2025-10-25

**Soundness:** 1
**Presentation:** 3
**Contribution:** 2
**Rating:** 2
**Confidence:** 4

**Summary:**

This paper introduces two extensions to existing formal explanation frameworks in the XAI literature: (1) “$\delta$-robust abductive explanations” (AXPs) generalize prior definitions by considering robustness within $\delta$-neighborhood. This concept is closely related to *inflated explanations* [1], but differs by avoiding the maximization over perturbation radii and instead restricting attention to a fixed $\delta$-surrounding. (2) The authors further propose *group explanations*, termed “g-$\delta$-AXPs”, which extend the $\delta$-robust AXP definition to sets of inputs rather than single instances. The approach is evaluated on several small-scale tabular benchmark datasets, one synthetic dataset, and small neural networks, and is implemented using the Marabou neural network verifier. The experiments focus on three main aspects: (1) Demonstrating that $\delta$-robust AXPs can be computed faster than inflated explanations, as they avoid exploring the largest possible certified region (at the cost of covering a smaller portion of it). (2) Comparing g-$\delta$-AXPs to LIME, SHAP, and an aggregation baseline, showing that the proposed explanations better align with the FFA metric introduced in [2]. (3) Showing that computing $\delta$-robust group explanations is more efficient than computing explanations independently for each instance in the group.

[1] Delivering Inflated Explanations (Izza et al., AAAI 2024)

[2] On Formal Feature Attribution and its Approximation (Yu et al., arxiv 2023)

**Strengths:**

1. The paper introduces extensions to existing formal definitions in XAI, which may be of interest to researchers in the field.

2. The authors conduct experiments across a range of benchmarks, although they are lacking (see Weaknesses).

3. The proposed extension from single-input to group-based explanations could be useful in certain scenarios.

4. The comparison using the FFA metric offers a potentially interesting evaluation direction for AXP-based explanations, but its conceptual justification requires substantial strengthening (see Weaknesses).

**Weaknesses:**

1. **Incremental definitions**. The definitions introduced are largely incremental extensions of existing concepts. AXPs have already been thoroughly defined and studied in prior research, and the use of a $\delta$-neighborhood around the irrelevant (non-explanation) features is a well-established convention in the literature. The paper’s choice to apply perturbations over the explanation features rather than the non-explanation features was previously explored by Izza et al. through their formulation of "inflated explanations". The primary difference here is that the authors constrain the perturbation to a limited $\delta$-region rather than the full inflated region. Similarly, the introduction of "group" explanations appears to be a straightforward extension, as group-based reasoning is already widely used across many XAI methods, such as in counterfactual explanations (as the authors themselves point out). Also, the definition of "inflated explanations" should be formally defined within the paper since it is a central one.


2. **Use of the FFA metric**. A major component of the evaluation relies on the FFA score, yet this metric is neither formally defined nor sufficiently contextualized in the paper, despite playing a central role in the results. FFA is not a standard evaluation metric in XAI and originates from a relatively recent preprint. Moreover, it is conceptually questionable to use a metric rooted directly in the AXP framework to compare $\delta$-gAXPs against methods such as LIME and SHAP, which optimize entirely different explainability tasks (approximating a local surrogate model, or a game-theoretic notion). The authors state, for example, that “Table 1 shows that $\delta$-gAXPs are more accurate than SHAP and LIME,” but this claim is misleading: SHAP and LIME are not designed to optimize AXP-consistent explanations, so it is expected that they would perform poorly under an AXP-based evaluation metric. Developing principled evaluation metrics for explanation quality is indeed important, but the comparison here does not convincingly demonstrate that $\delta$-gAXPs produce inherently “better” explanations - only that they align with the metric used.


3. **Lack of qualitative evaluation**. The paper does not provide any qualitative examples illustrating how $\delta$-gAXPs yield more interpretable or useful explanations for humans compared to existing approaches. Without case studies or visualizations - even simple tabular or image-based examples - the usefulness of these explanations in practical or human-centered contexts remains unclear.


4. **Experiments could be improved** The authors claim that Marabou is the state-of-the-art tool for neural network verification, which is inaccurate - since $\alpha$-$\beta$-CROWN is. Additionally, the neural network models used in the experiments are notably small, even when compared to this area of research, raising questions about scalability. The paper also lacks justification for the specific architectures chosen: Are these models drawn from prior work? Do they correspond to common architectures? Without such justification, there is a risk of cherry-picking models that favor the proposed method.


5. **Limited coverage and expected efficiency gains** While $\delta$-explanations are reported to be faster to compute than inflated explanations, the trade-off in coverage is significant, achieving only 20–50% of the inflated explanation coverage. This seems like a substantial reduction, and its implications are not adequately discussed. Moreover, the reported efficiency gains - both for $\delta$ versus inflated explanations and for group versus individual certification - are largely expected and do not constitute surprising or particularly insightful findings.

**Questions:**

1. How is the inflated explanation (used as a baseline) actually computed in practice? Specifically, is the maximum perturbation region determined using binary search over the verifier, and is the resulting explanation only approximated up to some $\epsilon$ tolerance?

2. How would the results differ if the comparison used Anchors instead of LIME or SHAP? Since Anchors also produces rule-based explanations, this may be a more meaningful or fair baseline for comparison.

3. Are there alternative, more standard evaluation criteria from the XAI literature - such as faithfulness, fidelity, stability, or robustness - that could serve as comparison metrics instead of relying solely on the FFA score? Prior work offers several well-established measures of explanation stability that may better align with accepted XAI conventions.

4. What is the relationship between the definitions introduced in this paper and the recent notion of “space explanations” proposed by Labbaf et al. (TACAS 2025) [3]?

[3] Labbaf et al., Space Explanations of Neural Network Classification, TACAS 2025.

---

### Official Review · Reviewer_8NEa · 2025-10-28

**Soundness:** 2
**Presentation:** 4
**Contribution:** 2
**Rating:** 4
**Confidence:** 4

**Summary:**

The paper identifies a gap in the study of formal abductive explanations (AXp-s): existing works do not address the robustness of the features involved in an explanation. To bridge this gap, the authors introduce $\delta$-AXp, a variant of AXp that remains robust under perturbations of its features, along with an algorithm to compute it given a triplet $⟨\delta$-perturbation, input sample, network$⟩$. They further generalize this concept to $\delta$-gAXp, which ensures robustness for groups of features, and propose an efficient algorithm for its extraction.

The proposed algorithms are evaluated on a fully connected network trained across multiple (real and synthetic) classical machine-learning benchmarks, using Marabou as the underlying verifier (implementing the “GUARANTEED” function in Algorithm 1). The authors demonstrate the value of $\delta$-AXp by analyzing the trade-off between computation time and coverage (Figure 4) in comparison with inflated AXp [Izza 2024b]. Moreover, by comparing the feature-importance error and correlation of $\delta$-gAXp with SHAP, LIME, and aggregated-group FFA relative to the average FFA score per sample, they show both superior performance (Table 1) and relative effectiveness (Figure 5).

**Strengths:**

1. Readability: The presentation is very clear.
2. Motivation: The paper identifies a gap in formal XAI for real-world applications.
3. Solution: The proposed $\delta$-AXp provides a simple and effective solution to address this problem.
4. Evaluation: The authors made a solid effort to compare their method with other approaches using established metrics.

**Weaknesses:**

1. Limited Evaluation:
To meet the comprehensiveness standards for ICLR, at least 2-3 of the following aspects should be improved:
- Scalability: The experiments do not demonstrate scalability — only a single fully connected model with two hidden layers of 10 neurons each is tested. Larger networks and/or other architectures should be evaluated.
- Extendability: The experiments do not show whether $\delta$-AXp and $\delta$-gAXp can be applied to regression tasks or contrastive explanations.
- Simple benchmarks: The use of classic ML benchmarks does not justify the need for explaining DNNs; other (more explainable) ML tools can be used as well.
- Verification diversity: Using a single verification tool is acceptable but represents the minimal evaluation setup.
2. Limited Novelty: The theoretical contribution is limited and does not include novel analyses or deep insights.

3. Simplifying Assumptions:
- The limitations (which are clearly mentioned in the paper) significantly weaken the supporting evidence and decrease the soundness. In addition, the following assumptions are too simplified:
- "we only show data for values of $\delta$ where at least 50% of the samples have a $\delta$-robust AXp" (line 348).
- For $\delta$-gAXp, the authors assume that the samples are sufficiently close to each other. This assumption is not supported by a probabilistic analysis or other justification.

**Questions:**

1. $\delta$-AXps have no formal guarantees, as mentioned in the Limitations Section. Instead of using the variance value, the distance of the farthest point from the mean should be used, formally guaranteeing $\delta$-robustness for all input samples while preserving performance.
2. The contributions of $\delta$-AXp and $\delta$-gAXp are orthogonal and not inherently linked. Can other XAI methods (e.g., $\epsilon$-AXp) be generalized in a similar way to how $\delta$-AXp is extended to $\delta$-gAXp?
3. Algorithm 1: By definition (line 152), a $\delta$-AXp is a pair $(E,\delta)$. Alg. 1 minimizes $E$ for a given value of $\delta$, but we might also want to maximize $\delta$ for a given $E$. More specifically, the pair $\langle E,\delta\rangle$ returned by Alg. 1 can often be improved by increasing $\delta$, making the result only a local optimum. Addressing this could improve comparison to Inflated Abductive Explanations [Izza et al., 2024b].
4. Figure 2: The range of the feature “$K$” (left side) is not visible.
5. The evaluated model is very small. Does the method have scalability issues?
6. Table 1: It seems that $\delta=0.00$ and $\delta=0.02$ most often yield the best results. Why?
7. Figure 3: It is unclear why the right-hand explanation is better. Could the authors clarify this? Also, was MNIST used for experiments? No details are provided.
8. The main goal of Alg. 1 is to find a minimal $E$ for a given $\delta$, but no comparison of the size of $E$ is presented.
9. How often does Alg. 2 (line 3) return False in the experiments?

Typos:
- Line 187: “otherwise, include…” (remove “include”)
- Alg 2, line 5: x → \bar{x}

---

### Official Review · Reviewer_Rg8d · 2025-11-07

**Soundness:** 1
**Presentation:** 3
**Contribution:** 2
**Rating:** 2
**Confidence:** 4

**Summary:**

This paper introduces two extensions to the concept of formal abductive explanations (AXps) for machine learning models. First, it proposes $\delta$-robust AXps, a formulation where an explanation remains valid not just for a single data point, but for an interval of size delta around the specified feature values. This aims to address the issue of traditional AXps to minor input perturbations. Second, it defines $\delta$-group AXps ($\delta$-gAXps), which seek to provide a common explanation for a group of similar samples by computing a $\delta$-robust AXp on a representative sample (the group mean), with delta scaled by the feature-wise standard deviation. The authors provide algorithms for these methods and evaluate them on neural networks using the Marabou verifier. They claim their approach is computationally cheaper than the related work of inflated AXps and that $\delta$-gAXps effectively capture group-level feature importance.

**Strengths:**

- The paper is very well written and logically structured. Examples, figures, the descriptions of the method's algorithms, and the provided source code further highlight its scientific contribution and support the authors' idea.
- The work addresses two important limitations of existing formal explanation methods: their fragility to small perturbations and their focus on individual instances. The motivation for developing robust and group-level explanations is compelling and has clear relevance.
- The related literature is thoroughly presented, clearly placing this work's contributions within the broader landscape of both heuristic and formal explainable AI. Additionally, there is an extensive discussion and comparison with the similar approach of inflated abductive explanations in sec. 3.1.1.

**Weaknesses:**

Despite the clear presentation and important problem motivation, the work suffers from significant weaknesses regarding its novelty, methodological assumptions, and the rigor of its experimental evaluation. I am willing to discuss the following points with the authors and other reviewers:

- The core notion of $\delta$-robustness for explanation affected by perturbations is not totally novel (see e.g., Fokkema et al. JMLR 2023 https://www.jmlr.org/papers/v24/23-0042.html or Lakkaraju ICML 2020 https://proceedings.mlr.press/v119/lakkaraju20a.html).  While the authors apply this concept specifically to abductive explanations, the conceptual contribution is more incremental than foundational. The paper's framing (in particular up to section 3.1) should be modulated to more accurately reflect that its primary contribution lies in the specific algorithmic implementation and the extension to group settings, rather than the invention of $\delta$-robustness itself.
- The proposed method's reliance on a pre-specified $\mathbf{\delta}$ is a critical flaw that is not adequately addressed. This vector (conveniently set to all equal values) must be provided as a input. This does not solve the problem of finding a robust region but rather shifts the search burden from the algorithm (as in inflated AXps, which find maximal bounds) to the user, who must engage in a trial-and-error process to find a non-trivial $\delta$ for which an explanation even exists (this failure is acknowledged in multiple points in the paper).
- The comparison with the inflated AXp of section 3.1.1 (while being a strength and pertinent for this contribution) reports only their limitations, and it is not really fair. All trade-offs between the two similar methods should be discussed to give a balanced and unbiased opinion to the reader, which will then ideally choose the method that best suits their needs. Compared to the inflated APx, the  $\delta$-robust approach is symmetric and the bounds are sub-optimal, and require the definition of $\mathbf{\delta}$. Also, while inflated APx might return low bounds for the feature considered last, $\delta$-robust are order-dependent too (as briefly and qualitatively discussed in the Appendix) and may fail as well, requiring multiple calls to the oracle for each value of delta
- The methods of group AXps rely on features' means and standard deviations, which are a straightforward extension to groups but should be based on questionable statistical assumptions. The use of the group mean as a "representative" and the standard deviation to scale delta is only statistically sound if the underlying feature distributions are unimodal and roughly symmetric (i.e., Gaussian). This approach is ill-defined for categorical data (a critical omission, given the patient example used for motivation) and may produce misleading results for skewed or multi-modal continuous features. Also, Fig 3 seems inconclusive and hard to interpret: how is the last image for the group explanation qualitatively (or better quantitatively) different? Many green/relevant pixels are on the boundaries of the image, but they are arguably irrelevant for MNIST classification.
- The empirical evaluation lacks the standards of rigour and completeness expected for a venue like ICLR. Raw performances of the trained NN are missing. Explanations are as good as the model they try to explain, and this is not provided (it seems also lacking in the evaluate_performance.py). Furthermore, details on hyperparameter selection, the number of training runs, and whether the same architecture was used across all datasets are not reported, suggesting a potential lack of rigor in the modeling and training process itself.
- The entire evaluation of the group explanation method hinges on the Formal Feature Attribution (FFA) metric from Yu et al. (2023), which is referenced to an arXiv preprint, not a peer-reviewed work, and that appears to have been rejected from at least one major venue (TMLR https://openreview.net/forum?id=JESCxuSMmi). Building a core evaluation framework upon such an unvetted foundation is a major methodological weakness (unless properly acknowledged and discussed).

Minor comments that did not affect the score:
-Use of LLMs section, required by the venue, is missing.
- $\mathbb{F}$ as a set of features in the AXp definition in Sec 3 is not formally defined, although it is clear what $E$ is.
- In lines 188-189, the set $E$ is written in standard form as E.
- For clarity, Algorithm 1 should also contain the required inputs (delta,x, and the oracle/GUARANTEED) as the first line (similarly to line 182). Also, in its description, the GUARANTEE function is described in lines 196 of the paper, but it should be mentioned earlier, as it is used in the second line of the algorithm. Similarly for Algorithm 2.
- Reference in the Appendix should be without parentheses.
- This work seems related to "Categorical Explaining Functors: Ensuring Coherence in Logical Explanations" https://doi.org/10.24963/kr.2025/30, which frames $\delta$-robustness in terms of formal logic. Although this is contemporaneous with the conference submission, it may be interesting for the authors.

**Questions:**

- Could you describe the practical workflow for a user to select an appropriate $\delta$? Given that the algorithm can fail if it is too large, doesn't this necessitate an iterative, computationally expensive search for it, undermining the claimed runtime advantages over inflated AXps (which takes a bunch of seconds)?
- What is the rationale behind choosing Marabou and not other verification methods? This point is central in your implementation and should be motivated further. Does the performance or applicability of your framework depend on specific features of Marabou?
- Could you provide a fairer comparison of runtime? Specifically, what is the runtime of your method to find a $\delta$-robust AXp that covers an equivalent volume to that found by an inflated AXp? How sensitive are the results for inflated AXps to the precision parameter $\eta$, and why was $\eta=0.01$ chosen? This is a crucial comparison that is missing, since you stated that "Computing an inflated AXp is more computationally demanding than computing a δ-robust AXp, since to inflate, an oracle needs to be called potentially hundreds of times," but do not account that the volume is also larger. How can the volume be smaller if $\delta>\eta$? How does inflated AXp compare on the 50% of the data that you excluded? What are the units of Figure 4, right?
- Can you provide an example of a "real-time decision support systems or in applications with limited computational resources, where they aim to find explanations for a large, but not necessarily maximal, feature intervals.", for which your model is aimed? How do you propose to extend the group methodology beyond its reliance on mean/std. deviation to handle categorical features or data with highly non-Gaussian distributions, which are common in real-world applications?

---

### Author Response · Authors · 2025-12-04

Hi all,

We thank all of the reviewers for the detailed and thoughtful reviews! Each of them are quite useful for us. We feel that some of the criticism is a matter of argument and presentation on our side, but many fair critiques were also pointed out. Due to the overall negative score, we will withdraw the paper. We will take the comprehensive comments presented in the reviews into account in future iterations of our work.

---

### Note · Authors · 2025-12-04

I have read and agree with the venue's withdrawal policy on behalf of myself and my co-authors.